# A Low-Cost, Wireless, 3-D-Printed Custom Armband for sEMG Hand Gesture Recognition

**DOI:** 10.3390/s19122811

**Published:** 2019-06-24

**Authors:** Ulysse Côté-Allard, Gabriel Gagnon-Turcotte, François Laviolette, Benoit Gosselin

**Affiliations:** 1Department of Computer and Electrical Engineering, Université Laval, 1065 Avenue de la Médecine, Quebec, QC G1V 0A6, Canada; benoit.gosselin@gel.ulaval.ca; 2Department of Computer Science and Software Engineering, Université Laval, 1065 Avenue de la Médecine, Quebec, QC G1V 0A6, Canada; francois.laviolette@ift.ulaval.ca

**Keywords:** acquisition system, surface electromyogram, sEMG, wearable sensors, gesture recognition

## Abstract

Wearable technology can be employed to elevate the abilities of humans to perform demanding and complex tasks more efficiently. Armbands capable of surface electromyography (sEMG) are attractive and noninvasive devices from which human intent can be derived by leveraging machine learning. However, the sEMG acquisition systems currently available tend to be prohibitively costly for personal use or sacrifice wearability or signal quality to be more affordable. This work introduces the 3DC Armband designed by the *Biomedical Microsystems Laboratory* in Laval University; a wireless, 10-channel, 1000 sps, dry-electrode, low-cost (∼150 USD) myoelectric armband that also includes a 9-axis inertial measurement unit. The proposed system is compared with the Myo Armband by Thalmic Labs, one of the most popular sEMG acquisition systems. The comparison is made by employing a new offline dataset featuring 22 able-bodied participants performing eleven hand/wrist gestures while wearing the two armbands simultaneously. The 3DC Armband systematically and significantly (p<0.05) outperforms the Myo Armband, with three different classifiers employing three different input modalities when using ten seconds or more of training data per gesture. This new dataset, alongside the source code, Altium project and 3-D models are made readily available for download within a Github repository.

## 1. Introduction

The way people interface with machines is constantly evolving with the aim of bridging the gap between human intention and machine action. Improved interfaces can profoundly alter the way entertainment is consumed or even change lives by elevating the autonomy of people living with disabilities. In certain situations, physical interfaces (e.g., touch screen and keyboard) can be replaced with the conscious modulation of biological signals by the user.

In the context of upper-limb amputees, the signals provided by muscular activity offer an attractive modality from which a user’s intention can be derived. Surface electromyography (sEMG) can also be leveraged to achieve intuitive interfaces in a vast array of domains for able-bodied participants [1,2,3]. sEMG signals are non-stationary and represent the sum of subcutaneous motor action potentials generated through muscular contraction [4]. In contrast with intramuscular EMG signals, which are recorded using needles that penetrate the muscle, sEMG signals are recorded directly on the participant’s skin surface [1]. While the latter has the advantage of being noninvasive, important noise is introduced when going further away from the muscle fibers, especially when nonintrusive dry electrodes are employed instead of their more intrusive and robust gel-based counterpart [5]. The quality of the sEMG acquisition system is thus crucial in obtaining as clear of a signal as possible. However, current systems tend to be expensive, often costing several thousands of dollars per channel or making noticeable compromises on the quality of the recorded signal (see Section 2 for details).

As such, the main contribution of this work is to present a new 3-D-printed wireless sEMG acquisition system based on an application-specific integrated circuit (ASIC). This armband aims at providing state-of-the-art recording while being low cost to produce and small enough to be easily wearable. The proposed device is referred to in this work as the 3DC Armband and was designed by the *Biomedical Microsystems Laboratory* in Laval University (BML-UL). Additionally, while comparisons of different sEMG acquisition system have been done in the past [6], these comparisons are made across different datasets and are thus generated from different conditions. Other works have explored the impact of sampling rate on gesture classification accuracy [7], but they did so by downsampling the signal of an acquisition system. While this allows for a direct comparison of the impact of the signal bandwidth on classification performance, such comparisons do not take into account the technical limitations associated with a higher bandwidth when building sEMG acquisition systems. To the best of the authors’ knowledge, this is the first time that a direct comparison between two different armbands cadenced at different sampling rates (200 vs. 1000 sps)is made on the same dataset. Finally, an additional contribution of this work is the publication of a new dataset recorded with both the Myo Armband and the proposed acquisition system on 22 able-bodied participants for eleven hand/wrist gestures. The dataset, alongside the 3-D models, Altium project, and the source code used in this article, are made readily available to the community (https://github.com/UlysseCoteAllard/3DC-Armband).

This paper is organized as follows. An overview of different sEMG acquisition systems is given in Section 2. Section 3 presents the proposed 3-D-printed sEMG armband in detail. The new dataset is detailed in Section 4. The methods employed for the comparison between the Myo armband and the proposed system are detailed in Section 5, and the results of this comparison are presented in Section 6. Finally, this work’s outcomes and possible improvements are discussed in Section 7.

## 2. Overview of Surface EMG Acquisition Systems

By their nature, sEMG signals are recorded with multiple layers of material between the electrode site and the muscle fibers generating the signal. As such, particularly robust acquisition systems have been developed over the years to contend with the different types of contaminants associated with such signals (e.g., power line interference, motion artifact, and biosignal crosstalk) [8].

One such system employed for clinical research is the Ultium EMG by Noraxon systems [9] which can record up to 32 channels simultaneously at a rate of ∼2 ksps and a baseline noise of less than 1 μV. Each channel is fully self-contained and wireless, allowing researchers to target multiple recording sites. Each module also integrates a 9-axis Inertial Measurement Unit (IMU) sensor. Similar systems such as the Trigno Avanty by Delsys Systems [10] and DataLITE sEMG by Biometrics [11] are also available. While these systems are highly accurate, they necessitate preparation of the recording site (i.e., washing and sometimes shaving the subject’s skin) before fixing each module to the skin, often using medical tape. This, coupled with their high cost ranging between ∼$17,000–20,000 USD, often renders such systems impractical for consumer-grade applications.

In 2015, the Myo Armband by Thalmic Labs [12] was released as a new consumer-grade sEMG acquisition system. This wireless armband offering eight channels was retailed for several orders of magnitude less than medical-grade acquisition systems ($200 USD). The armband is also nonintrusive, requiring no preparation of the recording site of any sort. However, to attain this, concessions were made both in terms of data quality and signal bandwidth. Most notably, the armband is limited to a sampling rate of 200 sps with 8-bit precision and comprised of only 8 channels. Regardless of these limitations, the Myo Armband has been widely utilized in a wide array of research topics (e.g., robotic arm control [2], video game control [13], motor imagery [14], and sign language recognition [15]).

More recently, the gForce-Pro from Oymotion [16] was released. The armband is sampled at 1000 sps, enabling it to leverage the full spectra of the sEMG signal [7]. However, this sampling rate increase made the gForce-Pro six times more expensive than the Myo armband for the same amount of channels and recording resolution.

Several sEMG acquisition systems have been presented in the literature such as in Refs. [17,18,19,20,21]. However, these systems tend to not offer a fully developed wearable form factor [19] or are simply too bulky to be embedded within an armband [17,18,20,21].

A technical comparison of the main sEMG acquisition systems previously mentioned alongside the proposed 3DC Armband is given in Table 1. Note that, for the rest of this work, the Myo Armband will be used for comparison with the proposed sEMG acquisition system. The system by Thalmic Labs was selected as it is “arguably the most widely known EMG armband in research” [7] (mentioned in more than 1250 articles on Google scholar). Additionally, its price range is in the same order of magnitude as the one estimated for the proposed 3DC armband.

## 3. The 3DC Armband (Prototype)

The 3DC Armband, which is depicted in Figure 1, features ten sEMG recording channels cadenced at 1000 sps alongside a 9-axis Inertial Measurement Unit (IMU). The proposed armband weighs 63 g and is assembled with a custom System-on-Chip (SoC), featuring competitive performance for sEMG recording: input referred noise of 2.2 μV${}_{\mathrm{rms}}$, resolution of 10 bits, dynamic range of 6 mV${}_{\mathrm{pp}}$, and a bandwidth of 20–500 Hz. The 3DC Armband consists of two interconnected parts. The first is the sensor printed circuit board (PCB) that includes all the electronic components for multichannel sEMG signal conditioning and multichannel sEMG data acquisition through a custom ASIC, IMU data acquisition, and wireless data transmission. The second part is the armband receptacles holding the sEMG electrodes. Both parts are interconnected using a detachable Molex connector, enabling easy electronic and software updates outside the armband. The following subsections detail each component of the proposed sEMG acquisition system.

### 3.1. System Overview

The proposed sensor, of which the system-level concept is shown in Figure 2, consists of six main building blocks:A custom 0.13-μm complementary metal oxide semi-conductor (CMOS) mixed signal (i.e., analog and numeric circuits on the same die) SoC that can record 10 sEMG channels in parallel [22,23].An ICM-20948 low-power 9-axis IMU from InvenSense, USA. This component has a 3-axis gyroscope, a 3-axis accelerometer, and a 3-axis magnetometer.An nRF24L01+ low-power wireless transceiver from Nordic Semiconductors, Norway, which sends the sEMG and IMU data to a base station at a 1 Mbps datarate.An MSP430F5328 low-power microcontroller unit (MCU) from Texas Instruments, USA. This MCU is mainly used for interfacing the SoC, the IMU, and the wireless transceiver.The power management unit (PMU), which includes a 1.9-V low-dropout regulator (LDO) for powering the MCU, the wireless transceiver, and the IMU. The highest voltage in the sEMG sensor is 1.9-V, which is optimized for low-power consumption since it is the smallest viable voltage for powering the MCU, the IMU, and the wireless transceiver, yielding around half the power consumption compared with a typical 3.3-V power supply. The PMU also includes a 1.2-V LDO for powering the SoC, which is the recommend supply voltage for the 0.13-μm technology used in the SoC. The system is powered with a 100-mAh LiPo battery.The Molex connector (# 0529910308) used for connecting with the Armband and for programming the MCU.

Note that the SoC was originally developed for neural electrophysiological signal acquisition and is detailed in previous publications [22,23]. Additionally, this SoC was shown to be able to successfully acquisition sEMG data [24].

The complete sensor is shown in Figure 3a with the main building blocks identified. The sensor has a flexible part, allowing it to fold the rigid parts on top of each other to save space (See Figure 3c). When folded, the PCB occupies 1.25 cm${}^{3}$. Finally, the 3DC sensor communicates with a custom-based base station consisting of (i) an nRF24L01+ low-power wireless transceiver from Nordic Semiconductors, (ii) an ARM cortex M4 MCU from Texas Instruments for managing the data, and (iii) an FT232RL UART-to-USB chip from FTDI, United Kingdom, for sending the sEMG data to the computer. The following Section 3.2 and Section 3.3 give more details about the SoC, MCU firmware, and the IMU, while Section 3.5 presents the 3-D models that contain the armband’s electronics.

### 3.2. sEMG Acquisition Interface

Each recording channel of the SoC encompasses a low-noise and low-power fully differential bioamplifier, followed by a fully differential third-order Delta-Sigma (ΔΣ) multi-stage noise shaping (MASH) analog-to-digital converter (ADC) and an on-chip fourth-order cascaded integrator-comb (CIC) decimation filter [22,23]. The use of fully differential topologies (amplifier and ADC) doubles the dynamic range of the SoC (6-dB increase) while being more robust to external noise sources compared to a single-ended solution [25]. In this design, the bioamplifier is a single stage AC-coupled operational transconductance amplifier (OTA) which rejects any DC offsets generated at the electrode–skin interface. This topology also features pseudo-resistors in the feedback path to produce a high-pass analog cutoff frequency of ∼1 Hz.

Conventional ADCs require strict analog anti-aliasing filtering. This analog filtering is commonly performed with 2–3rd or higher order filtering, which increases the bioamplifier’s complexity, size, and power consumption. One advantage of using a ΔΣ ADC is to relax the constraints on the analog anti-aliasing filter. The proposed system performs an implicit fourth-order CIC anti-aliasing digital filter before decimation. Indeed, the ΔΣ pushes the *Nyquist* frequency at $F_{decim}\times OSR/2$, where $F_{decim}$ is the sampling frequency after decimation and OSR is the oversampling ratio; thus, less restrictive filtering between $F_{decim}/2$ and $F_{decim}\times OSR/2$ is required to avoid aliasing with this type of ADC.

As it can be seen in Figure 4a, the oversampling of the ΔΣ pushes the *Nyquist* frequency far from the bandwidth after decimation. For this application, an oversampling ratio (OSR) of 50 is employed to achieve an effective number of bits (ENOB) of 10 bits, pushing the *Nyquist* frequency to 25 kHz. The analog low-pass filtering is performed implicitly by the internal analog G-mC filter of the OTA [22,23] inside the bioamplifier, which cuts at −3 dB at ∼7 kHz (black curve in Figure 4a), leading to almost no aliasing, as there is a −12-dB attenuation at 25 kHz. The final low-pass filtering is performed by the fourth-order CIC decimation filter, which has a −3-dB low-pass cutoff frequency of 460 Hz, which is close to the ideal cutoff frequency of 500 Hz, and with a −80-dB attenuation per decade before the signal is downsampled to 1 kHz (blue curve in Figure 4a). Figure 4a also illustrates the Myo bandwidth for comparison. As can be seen, only a small portion (<100 Hz) of the proposed sensor bandwidth is covered by the Myo. The bioamplifier noise spectrum over a 500 Hz bandwidth is shown in Figure 4b. The total input referred noise is of 2.2 μV${}_{\mathrm{rms}}$ (20–500 Hz), which is smaller than the quantifying step of the ADC (resolution of 7 μV). The SoC communicates with an external MCU using a dedicated serial peripheral interface (SPI) bus and using a custom protocol to extract the data from all the channels.

The custom SoC employed for sEMG acquisition is wire-bonded onto a PCB substrate using 25 μm gold bonds and protected by an EPO-TEK 301 Glob-Top that was held in place during the curing phase using an AD1-10S dam from ChipQuick. An enlargement of the packaged SoC is depicted in Figure 3b.

### 3.3. MCU Firmware

The MSP430F5328 MCU controls the SoC, the IMU, and the wireless transceiver together by using three dedicated SPI busses. To use the least amount of hardware components as possible, the oversampling clock signal driving the SoC ΔΣs is provided by a pulse-width modulation (PWM) module within the MCU (set at 50% duty-cycle). The sEMG sampling is triggered by the PWM timer interruption when 50 clock cycles (OSR of 50, interruptions at 1 kHz) have been issued. Then, the MCU triggers one of the direct memory access (DMA) module channels to send commands to the SoC as to read all the 10 SoC channels one after the other. The acquired sEMG data is pushed automatically by the DMA within a First-in, First-out (FIFO) structure and sent to the wireless transceiver by another DMA channel when 20 or more bytes are available (10 × 2 bytes packets). Since the DMA module performs all the work, the MCU is idle most of the time. It is woken up at a 50-Hz frequency to get and forward the IMU data to the transceiver.

### 3.4. Inertial Measurement Unit

An IMU is a device consisting of accelerometers and gyroscopes from which the tracking of the device’s orientation can be derived. A tri-axis magnetometer can be added to form a hybrid IMU, sometimes referred to as a *Magnetic, Angular Rate, and Gravity* (MARG) sensor [26], to reduce the orientation accumulated error. Information from an IMU system is widely employed in the domain of rehabilitation [27]. Additionally, for dynamic sEMG-based gesture recognition, orientation information from IMU devices can be leveraged to obtain higher performances than with EMG alone [20,28,29]. Furthermore, IMUs have been employed to increase the number of gestures that can be detected by combining the orientation of the forearm with static hand gestures [30,31].

As the inclusion of an IMU device alongside sEMG channels allows a wider range of dynamic gestures to be detected, an armband featuring both modalities can be employed for a broader range of applications. Consequently, the ICM-20948, consisting of a low-power IMU featuring a 3-axis gyroscope, a 3-axis accelerometer, and a 3-axis magnetometer, was incorporated within the 3DC Armband. This IMU was selected for its small footprint (3 × 3 mm${}^{2}$ 24-QFN package), its low power supply capability (1.9 V supported), and its high resolution of 16 bits. This chip communicates with the MCU using a 4-wire serial peripheral interface (SPI) bus cadenced a 5 MHz. The MCU extracts three *<x, y, z>* vectors from the IMU for the accelerometer, gyroscope, and magnetometer by reading the first user bank of the IMU. This allows the MCU to read the data all at once (18 bytes block read). The vectors are then stored in a dedicated packet and sent to the wireless transceiver for further ex situ processing. The orientation data, in the form of a quaternion, is computed from the 9-axis IMU with the Madgwick’s algorithm [26] using the x-IMU implementation (https://github.com/xioTechnologies/Open-Source-AHRS-With-x-IMU) on the receiving computer.

### 3.5. 3-D Printing Models

The armband’s microelectronics are held by three different receptacles (shown in Figure 5) each fulfilling diverse functions. The system’s holder, depicted in Figure 5A, shows the system’s receptacle, which also houses the main electrode. The battery and a standard size electrode are stored in the battery holder, which can be seen in Figure 5B.

Finally, the eight remaining standard electrodes are housed in eight small electrode receptacles. The 3-D model of these receptacles is shown in Figure 5C. The hole in the top is there to facilitate the assembly of the system.

The circular holes on all three receptacles serve to pass elastic cords through, that link the different modules together and ensures that the armband can be worn by a wide variety of persons. The rectangular holes serve to pass small elastic bands through, on which the different electrical cables can be attached on to link the different microelectronic components together.

The overall armband price is valued at ∼$150 USD. The price was estimated using the ADS1298 from Texas Instrument as an estimation for the custom SoC and assuming the fabrication of 20 PCBs.

## 4. Comparison Dataset

Beyond the technical description presented in the previous section, the usefulness of the proposed armband must be assessed with real-life data. As such, a new dataset was recorded as to allow as close of a direct comparison as possibles between the Myo and the proposed 3DC Armband. The dataset is comprised of 22 able-bodied participants (7F/15M, 17/5 right/left handed) aged between 23 and 69 years old (average 34 ± 14 years old).

The data acquisition protocol was approved by the Comités d’Éthique de la Recherche avec des êtres humains de l’Université Laval (approbation number: 2017-0256 A-1/10-09-2018), and informed consent was obtained from all participants.

### 4.1. Data Acquisition Protocol

Before the recording started, both the Myo and the 3DC Armband were placed simultaneously on the dominant arm of the participant. The highest armband (i.e., the one closest to the elbow) was set to its maximum diameter and slid up until the armband’s circumference matched the participant’s forearm circumference. For the first participant, the Myo Armband was the one placed closest to the elbow. This process was replicated for each following participant but alternating the armband closest to the elbow between each subject. The two possible armband configurations alongside examples of the range of armband placements on participants’ forearm are shown in Figure 6. This method of positioning was adopted as to better represent the wide range of positions that nonexperts might use when wearing this type of hardware. The delay between putting the armband on the participant’s forearm and the start of the experiment was approximately three minutes on average.

The proposed dataset is made of eleven hand/wrist gestures, which are presented in Figure 7. All gesture recordings were made with the participants standing up with their forearm parallel to the floor supported by themselves. Starting from the neutral gesture, the participants were instructed, with an auditory cue, to hold the next gesture for 5 s. The cue given to the participants were in the following form: *Gesture X, 3, 2, 1, Go.* The recording of each movement began just before the movement was started and held by the participant as to capture the ramp-up segment of the muscle activity and always started with the neutral gesture. The recording of the eleven gestures for 5 s each totaled 55 s of data and is referred to as a *cycle*. A total of four *cycles* (220 s of data) were recorded with no interruption between cycles. Then, a five min pause was observed, where the participant could relax (without removing the armbands). After the pause, another four *cycles* of data were recorded. The first four *cycles* of data are referred to and serves as the *training dataset*, while the second group of *cycles* is referred to and serves as the *test dataset*. Note that the ramp-up period is included in the labeled dataset for each gesture.

### 4.2. Preprocessing

As the main use-case of the proposed armband is a real-time classification, a critical factor to consider is the input latency. The optimal latency (taking into account both classification performance and controller delay) was shown to be between 150–250 ms [32]. As the Myo Armband is limited to a sampling rate of 200 sps, a window size of 250 ms was selected as to not unduly give advantage to the proposed armband. Note that while the preprocessing is made so that it could be implemented for real-time classification, all results presented in this article are computed offline.

As mentioned in Section 3, the proposed 3DC Armband is band-pass filtered between 20–500 Hz. However, to produce a dataset as close as possible to the raw sEMG signal, the high-pass filter was instead set at ∼1 Hz using the SoC bioamplifier tunable pseudo-resistor bank. As such, the preprocessing of the dataset involves a fourth-order butterworth high-pass filter at 20 Hz as suggested in Ref. [7] for both armbands. An example of the signals recorded from both armbands after filtering is given in Figure 8.

## 5. Comparison Methods

The dataset previously described (Section 4) is employed to qualitatively discriminate between the Myo and 3DC Armband. The comparison is rendered from three different input modalities: a baseline feature set, the raw sEMG signals, and the signals represented in the time–frequency domain. The remaining section describes the three classification methods in detail.

It should be noted that one of the goals of this work is to generate a comparison that is as fair as possible between the proposed armband and the Myo Armband. As such, several choices were made to achieve this goal, sometimes to the detriment of the classification accuracy. These choices are explicitly detailed below.

### 5.1. Baseline Method

The baseline method employs a feature set (Hudgins’ Time-Domain Feature Set [33]) and a classifier (Linear Discriminant Analysis [1,34]) widely used in the literature. Both are described in the following two subsections.

#### 5.1.1. Hudgins’ Time-Domain Feature Set (H-TD)

Historically, the literature on sEMG-based gesture recognition primarily centers on feature-set engineering as to characterize the sEMG signals in a discriminative way [4,35].

Among all the feature sets proposed in the literature, the most commonly employed one is probably H-TD [7]. This set is comprised of four features from the time domain and are relatively inexpensive to compute:Mean Absolute ValueZero CrossingSlope Sign ChangesWaveform Length

Detailed descriptions of these four features are given in Ref. [36]. H-TD often serves as the basis for bigger feature sets [4,7]. As such, it is particularly well-suited as a baseline comparison between the Myo and the proposed armband.

#### 5.1.2. Linear Discriminant Analysis

Several types of classifiers have been employed in the past for sEMG-based gesture recognition. Some of the most commonly employed are the Support Vector Machine (SVM) [7], Artificial Neural Networks (ANN) [4], and Linear Discriminant Analysis (LDA) [1,34].

The latter is widely employed in the domain as it is a timely and computationally efficient classification technique both at training and prediction time while still being able to achieve high classification accuracies [1,37].

While, SVM with has been shown in some work to be able to achieve higher classification accuracies than LDA [7], it requires hyperparameter optimization, which could bias the results towards one specific armband. On the contrary, LDA does not require any hyperparameter optimization and can thus be employed to compare the armbands more fairly.

### 5.2. Raw sEMG Classification

With the recent advent of deep learning, the raw sEMG signal can be employed directly for gesture classification [36,38], something which was considered “impractical" before [4].

The raw data is passed as an image of shape Channels X Samples (i.e., 8×50 for the Myo Armband and 10×250 for the 3DC Armband) to a ConvNet. Note that the raw signal is first band-pass filtered as described in Section 4.2. The ConvNet architecture, which can be seen in Figure 9, is based on the one presented in Ref. [36] as it was shown to be comparable to the current state-of-the-art. The main difference between the two is the use of Global Average Pooling in lieu of the fully connected layer to reduce the number of parameters.

The architecture used is the same for all armband configurations to not overly give advantage to one over the other. Adam [39] is employed for the ConvNet’s optimization with an initial learning rate of 0.0404709 (as used in Ref. [36]). Learning rate annealing is applied with a factor of five and a patience of five epochs. Training is done with batch size of 512, and the dropout rate is set at 0.5.

### 5.3. Time-Frequency Domain Classification

As to better investigate the potential impact of the greater sampling frequency of the 3DC Armband, the classification performance using features of the time-frequency domain is investigated.

Similar to References [36,40], the short-time Fourier transform-based spectrograms are considered for the characterization of the sEMG signals in the time-frequency domain. For both armbands, the spectrograms are computed with Hann windows of 100 ms and an overlap of 50 ms. These hyperparameters were chosen using the *training dataset* and to use a similar ConvNet architecture for both armbands. As suggested in References [36,40] appropriate axis swaps are applied to yield a final image of 4×8×11 and 4×10×51 (i.e., time × channel × frequency bins) for the Myo and 3DC Armband respectively. This example’s formatting allows the convolutions to be performed on spatial X frequency information, while the time is considered as different viewpoints of the same event.

The example is then fed to the ConvNet represented in Figure 10. Except for a learning rate of 0.00681292 (as used in Ref. [36]), all hyperparameters are as described in Section 5.2.

## 6. Results

All results given in this section are computed from all four *cycles* of the *test dataset*. Training of each classifier is done with one, two, three, and four training *cycles* (i.e., 5, 10, 15, and 20 s of training data per gesture respectively). Additionally, due to the stochastic nature of the deep learning-based algorithms considered in this work, all results from each participant for each amount of training *cycles* are given as an average of 20 runs.

For statistical analysis purposes, each participant is considered as a separated dataset. As suggested in Ref. [41], the Wilcoxon Signed Rank test [42] (n=22) is applied to compare between the Myo and 3DC Armband.

The comparison between the armbands with the LDA classifier is shown in Figure 11, while the confusion matrices for four cycles of training with the LDA classifier are given in Figure 12. Similarly, the comparison and associated confusion matrices for the *RAW* and *Spectrogram*-based classifiers are given in Figure 13, Figure 14, Figure 15 and Figure 16 respectively. The rest of the training cycles’ confusion matrices are shown in Appendix A.

## 7. Discussion

Figure 11, Figure 13, and Figure 15 show that, in all cases, the proposed armband outperforms the Myo Armband. This difference is judged significant for all instances involving two or more cycles of training by the two-tailed Wilcoxon signed-rank test. As expected, augmenting the amount of training examples systematically improves the performance of all tested classifiers for both armbands, corroborating the results in Ref. [36].

The confusion matrices show that the hardest gestures to differentiates between for all three classifiers and both armbands are the Chuck and Pinch Grip. This is expected considering they only differ by the flexion of the middle finger. Additionally, the 3DC Armband tends to outperform the Myo across all gestures. As such, one could expect that the higher spatial dimension and frequency rate yield an advantage to the proposed armband that is not gesture-specific.

Overall, the *Raw Convnet* was the best performing classifier for both armbands, achieving an average accuracy of 89.47% and 86.41% for four cycles of training with the 3DC and Myo Armband respectively. For comparison, LDA obtained 84.81% and 80.00% for four cycles of training with the 3DC and Myo Armband respectively. While more sophisticated feature engineered sets exist [1,4], these results support previous findings showing the exciting potential of feature learning within the context of sEMG gesture recognition [36,43].

When comparing the performance of the Myo and 3DC Armband with a single training cycle, the difference in accuracy is not judged statistically significant for both the LDA and Raw ConvNet classifier. This might be due in part to the increased spatial and frequency information provided by the 3DC Armband which naturally increases within-class variability. However, another hypothesis could be that the warm-up period (i.e., the time between putting the armband on the participant’s forearm and the start of the experiment) was not long enough for the 3DC Armband [44]. Indeed, as the 3DC electrodes are half the surface area of the Myo’s, a greater area of the skin was in contact with the Myo Armband. Thus, our hypothesis is that the Myo requires less sweat and humidity per square centimeter between the electrode and the skin to achieve a good ionic conduction. To verify this hypothesis, all three classifiers were retrained with only the last cycle of training recorded for each participant. This provides the longest warm-up period possible on the training dataset for both armbands.

For all three classifiers, the proposed sEMG acquisition system again outperforms the Myo armband. However, this difference is judged significant by the two tailed Wilcoxon signed-rank test only for the LDA classifier (*p*-value = 0.0309). This seems to suggest that while a longer warm up period might help the 3DC Armband, it cannot, on its own, explain why the two systems perform similarly when employing only a single cycle of training.Consequently, the proposed warm-up period hypothesis cannot be confirmed with the available results. As such, it might be that, when very few examples are available for training, the increase in computational cost is not worth augmenting the spatial and frequency sampling rate resolution. It would be interesting to see how transfer learning algorithms developed for sEMG data affect these results [36,40].

Future works will focus on slightly enlarging the contact area provided by 3DC Armband while making sure to not design overly large electrodes which would increase the noise of the signal from crosstalk [19]. A potential added benefit of enlarging the contact area is reducing the effect of electrode shift [45]. The relationship between warm-up time and electrode size will also be characterized. Additionally, shielding will be incorporated between the inter-connections of the electrodes.

## 8. Conclusions

This paper presents a new wearable sEMG acquisition system. The 3-D-printed armband features 10 sEMG recording channels and is cadenced at 1000 sps. The whole system is light (63 g) and incorporates a 9-axis IMU and a custom SoC. This SoC features competitive performances for this application with an input referred noise of 2.2 μV${}_{\mathrm{rms}}$, resolution of 10 bits, dynamic range of 6 mV${}_{\mathrm{pp}}$, and a bandwidth of 20–500 Hz. The armband could be conceivably assembled for ∼$150 USD, making it more affordable and widely accessible than clinical-grade systems currently available.

The 3DC Armband was shown to significantly outperform the most widely used consumer-grade sEMG armband on a newly proposed dataset featuring 22 able-bodied participants performing 11 hand/wrist gestures.

Among the limitations of the proposed system is a possible longer warm up period than the Myo Armband. The relationship between electrode size and warm-up time will be investigated to provides a better skin-electrode interface. Shielding between the interconnections of the electrodes of the armband will also be added.

## Figures and Tables

**Figure 1 sensors-19-02811-f001:**
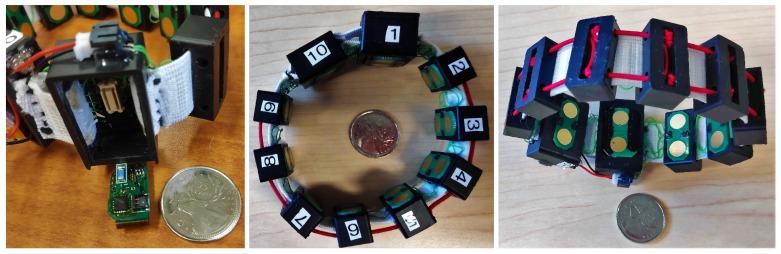
The proposed 3DC Armband. The system and the battery are held in the receptacles identified by 1 and 10 respectively. The label on each part of the armband corresponds to the channels’ order that are recorded for the dataset described in Section 4.

**Figure 2 sensors-19-02811-f002:**
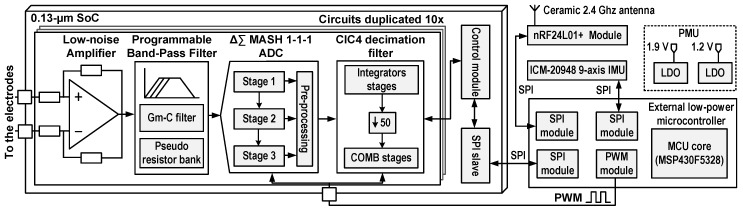
System-level concept of the multichannel wireless sEMG sensor: The sensor is built around a custom 0.13-μm SoC that includes 10× sEMG channels, each of which encompasses a bioamplifier, a ΔΣ analog-to-digital converter (ADC), and a 4th order decimation filter. The SoC, the nRF24L01+ low-power wireless transceiver, and the ICM-20948 9-axis IMU are interfaced with an MSP430F5328 low-power MCU.

**Figure 3 sensors-19-02811-f003:**
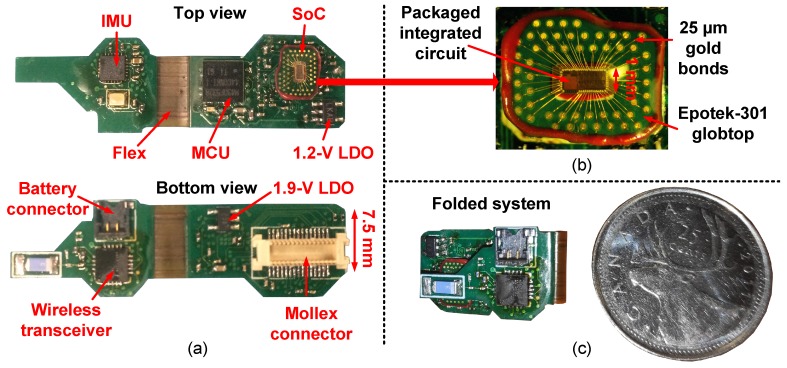
(**a**) Two-sided view of the sEMG sensor with each part identified: The printed circuit board (PCB) has a flexible region to fold the two rigid parts on top of each other to save space. (**b**) The packaged SoC which is wirebonded directly on a PCB substrate. (**c**) The system folded in its final position beside a Canadian quarter coin (diameter of 23.88 mm).

**Figure 4 sensors-19-02811-f004:**
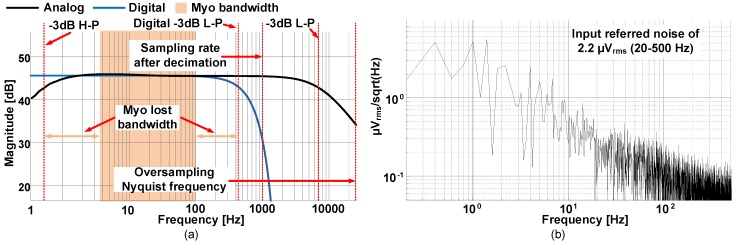
(**a**) Analog bandwidth of the bioamplifier (in black), digital bandwidth of the decimation filter (in blue), Myo bandwidth comparison (in orange), and (**b**) noise spectrum of the bioamplifier. The input referred noise is of 2.5 μV${}_{\mathrm{rms}}$ over a 500-Hz bandwidth.

**Figure 5 sensors-19-02811-f005:**
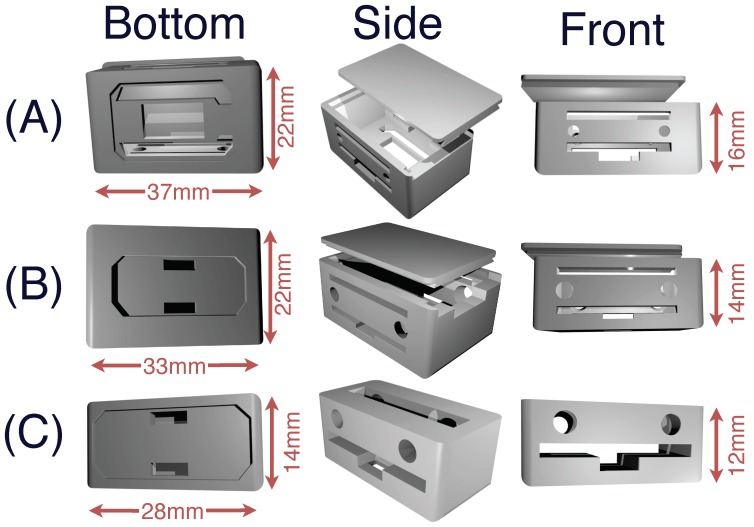
(**A**) The system’s receptacle: The bottom of the unit is used to receive the main electrode, while the system is stored inside. A cover slides on to enclose the system. (**B**) The battery holder: This receptacle is used to house the power source of the armband and, as such, should be placed next to the system’s holder. Once the battery is placed, the cover can then slide on to protect the system. A standard electrode is placed on the bottom of this holder. (**C**) This holder houses a standard electrode. For the proposed 3DC Armband, eight such receptacles are required.

**Figure 6 sensors-19-02811-f006:**
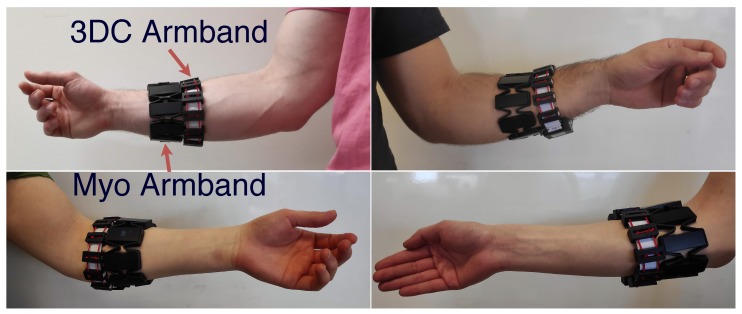
The two different armband configurations (left/right) employed in this work with the 3DC being either above or below the Myo armband with respect to the participant’s wrist. This figure also showcases the wide variety of armband positions recorded in the proposed dataset.

**Figure 7 sensors-19-02811-f007:**
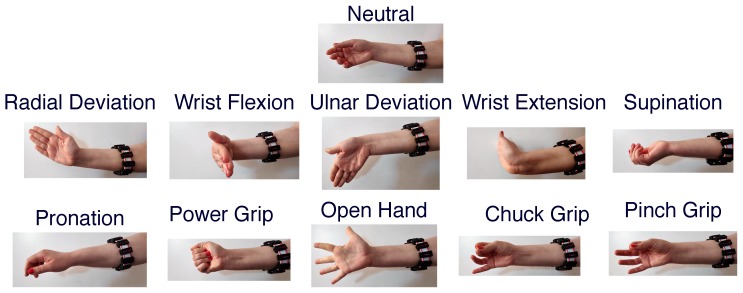
The eleven hand/wrist gestures employed in the proposed dataset.

**Figure 8 sensors-19-02811-f008:**
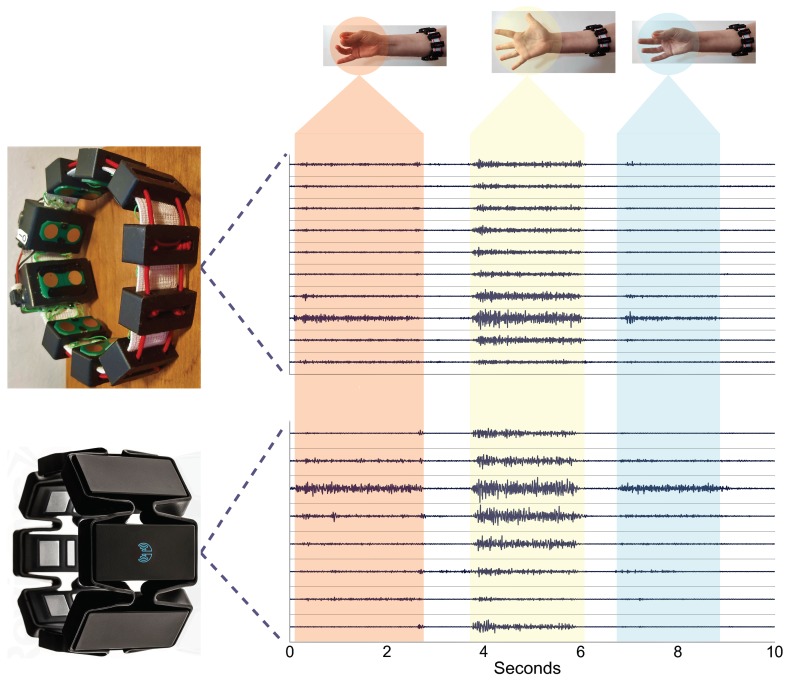
Comparison of the signals recorded with the Myo Armband and the proposed 3DC Armband. The *x*-axis represents time in seconds, while the *y*-axis is the different channels of the armbands. The three gestures recorded in order are the *chuck grip*, *Open Hand*, and *Pinch Grip*. Note that these signals were not obtained using the *Comparison Dataset* recording protocol to show a wider array of gestures in a continuous way.

**Figure 9 sensors-19-02811-f009:**
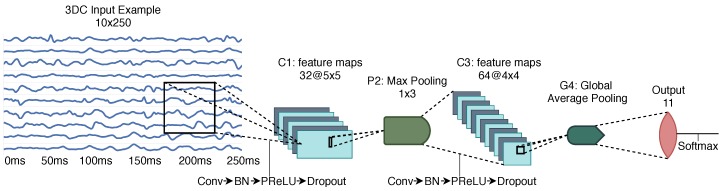
The raw ConvNet architecture employing 34,667 parameters. In this figure, *Conv* refers to *Convolution* and *BN* refers to Batch Normalization. While the input represented in this figure is that of the 3DC, the architecture remains the same for all considered systems.

**Figure 10 sensors-19-02811-f010:**
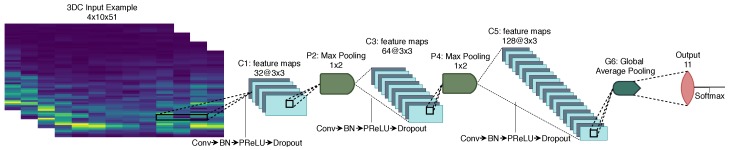
The Spectrogram ConvNet architecture employing 95,627 parameters. In this figure, *Conv* refers to *Convolution* and *BN* refers to Batch Normalization. The input represented comes from the 3DC Armband with the channels on the *x*-axis and the frequency bins on the *y*-axis. Due to the Myo Armband associated input size, P4 and C5 were removed from the architecture when training on Myo’s data.

**Figure 11 sensors-19-02811-f011:**
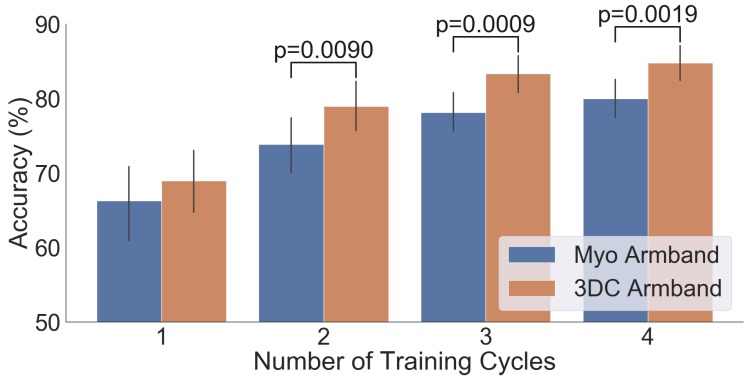
Comparison between the Myo and the 3DC Armband employing LDA for classification: The number of cycles corresponds to the amount of data employed for training (one *cycle* equals 5 s of data per gesture). The Wilcoxon Signed Rank test is applied between the Myo and the 3DC Armband. The null hypothesis is that the median difference between pairs of observations (i.e., accuracy from the same participant with the Myo or the 3DC Armband) is zero. The p-value is shown when the null hypothesis is rejected (significant level set at p=0.05). The black line represents the standard deviation calculated across all 22 participants.

**Figure 12 sensors-19-02811-f012:**
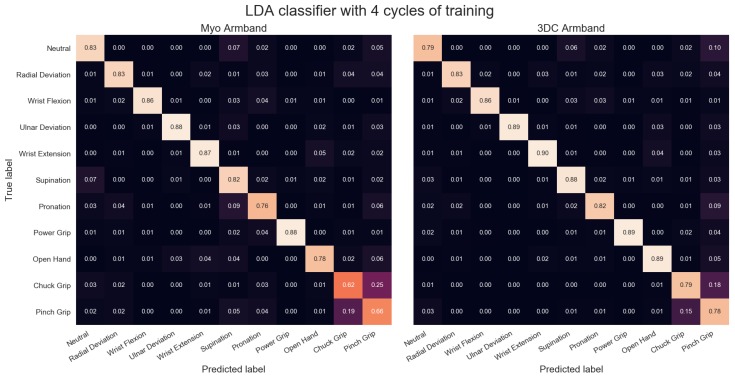
Confusion Matrices for the Myo and the 3DC Armband employing linear discriminant analysis (LDA) for classification and four cycles of training. A lighter color is better.

**Figure 13 sensors-19-02811-f013:**
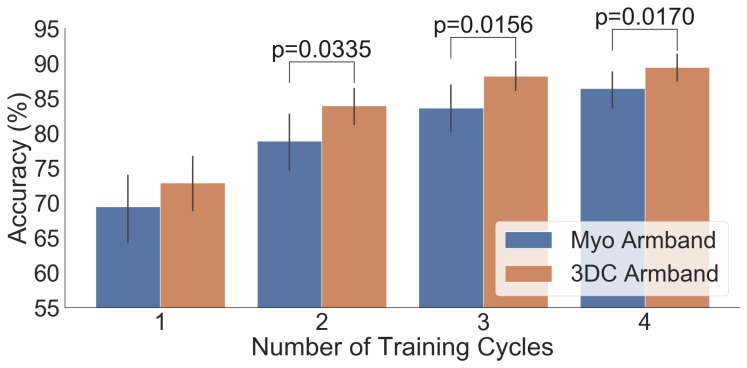
Comparison between the Myo and the 3DC Armband employing Raw ConvNet for classification: The number of cycles corresponds to the amount of data employed for training (one *cycle* equals 5 s of data per gesture). The Wilcoxon Signed Rank test is applied between the Myo and the 3DC Armband. The null hypothesis is that the median difference between pairs of observations (i.e., accuracy from the same participant with the Myo or the 3DC Armband) is zero. The p-value is shown when the null hypothesis is rejected (significant level set at p=0.05). The black line represents the standard deviation calculated across all 22 participants.

**Figure 14 sensors-19-02811-f014:**
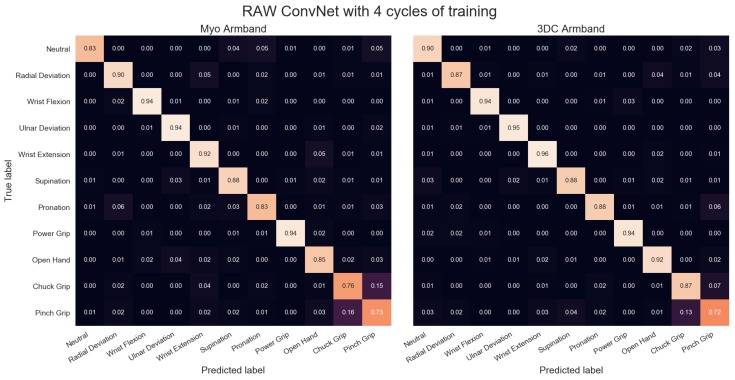
Confusion Matrices for the Myo and the 3DC Armband employing the *Raw* ConvNet for classification and four cycles of training. A lighter color is better.

**Figure 15 sensors-19-02811-f015:**
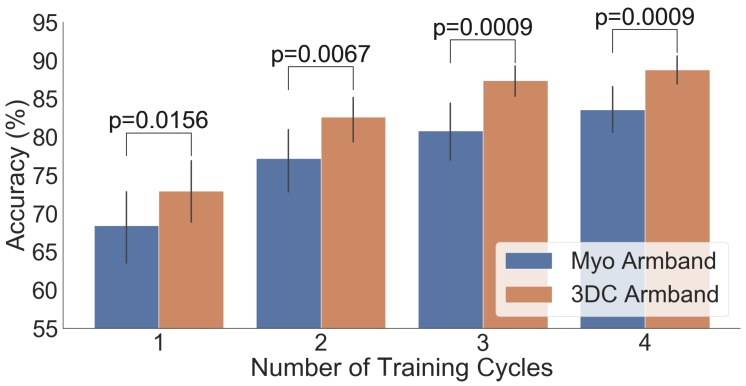
Comparison between the Myo and the 3DC Armband employing the *Spectrogram* ConvNet for classification: The number of cycles corresponds to the amount of data employed for training (one *cycle* equals 5 s of data per gesture). The Wilcoxon Signed Rank test is applied between the Myo and the 3DC Armband. The null hypothesis is that the median difference between pairs of observations (i.e., accuracy from the same participant with the Myo or the 3DC Armband) is zero. The p-value is shown when the null hypothesis is rejected (significant level set at p=0.05). The black line represents the standard deviation calculated across all 22 participants.

**Figure 16 sensors-19-02811-f016:**
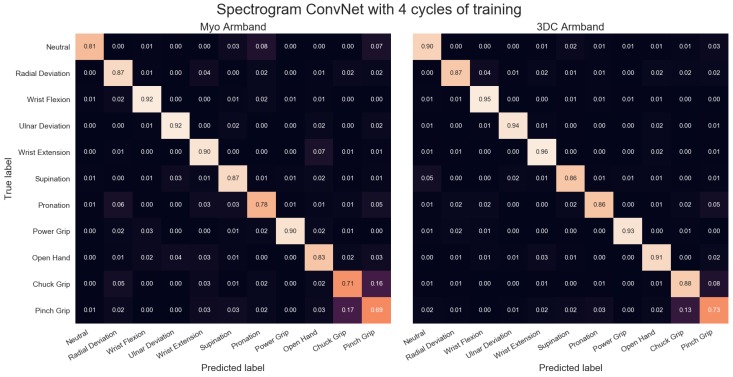
Confusion Matrices for the Myo and the 3DC Armband employing the *Spectrogram* ConvNet for classification and four cycles of training. A lighter color is better.

**Table 1 sensors-19-02811-t001:** Characterization of different surface electromyography (sEMG) acquisition systems. Values of N.A. mean that the information is not available. Note that, while the Hercules system [19] is included for completeness, it does not possess a wearable form factor.

	Delsys Systems Trigno Avanti	Biometrics DataLITE sEMG	Noraxon Ultium EMG	Oymotion gForce-Pro	Thalmic Lab Myo Armband	Hercules	3DC Armband
sEMG channels	up to 16	up to 16	up to 32 (at 2000 sps) or 16 (at 4000 sps)	8	8	8	10
sEMG ADC *	16 bits	13 bits	16 bits	8 bits	8 bits	12 bits	10 bits (ENOB *) (data sent on 16 bits)
sEMG Sampling rate	1960 sps	2000 sps	4000 sps	1000 sps	200 sps	1000 sps	1000 sps
Bandwidth or Built-in Filters	20–450 Hz or 10–850 Hz	10–490 Hz	5/10/20– 500/1000/1500 Hz	20–500 Hz	∼5–100 Hz	20–500 Hz	20–500 Hz
Contact Dimensions	5 mm${}^{2}$	78 mm${}^{2}$	N.A.	∼66 mm${}^{2}$	100 mm${}^{2}$	78 mm${}^{2}$	50 mm${}^{2}$
Contact Material	Silver	Stainless Steel	N.A.	Stainless steel silver coated	Stainless Steel	Gold plated Copper	Electroless nickel immersion gold (ENIG)
Full Scale (Peak to Peak)	+/−11 sps or +/−22 sps	+/−6 sps	+/−24 sps	N.A.	∼+/−1 sps (measured)	+/−6 sps	+/−3 sps
Input referred-noise (On system bandwith)	N.A.	<5μV	<1 μV	N.A.	N.A.	N.A.	2.2 μV
IMU * sensors	9-axis Acc, Gyro, Mag	No	9-axis Acc, Gyro, Mag (if EMG set at 2000 sps or below)	9-axis Acc, Gyro, Mag	9-axis Acc, Gyro, Mag	No	9-axis Acc, Gyro, Mag
IMU Sampling rate	24–470 Hz (Acc), 24–360 Hz (Gyro), 50 Hz (Mag)	-	200 Hz	50 Hz	50 Hz	-	50 Hz
Transmitter	BLE 4.2	WiFi	2.4 GHz	BLE 4.1	BLE 4.0	Wi-Fi	Enhanced Shockburst **
Autonomy	4 to 8 h	8 h	8 h	N.A.	16 h	N.A.	6 h
Weight	14 g (per channel)	17 g (per channel)	14 g (per channel)	80 g	93 g	N.A.	62 g
Price	∼$20,000 USD (for 16 channels)	∼$17,000 USD (for 16 channels)	∼$20,000 USD (for 16 channels and free battery replacement)	$1250 USD	$200 USD	N.A.	∼$150 USD ***

* ADC: Analog-to-digital converter; ENOB: effective number of bits; IMU: inertial measurement unit; BLE: Bluetooth low energy. ** 2.4 GHz low-power custom protocol (similar to BLE*) from Nordic Semiconductor, Norway. *** The cost of the System-on-Chip was replaced by the cost of a comparable product: the ADS1298 from Texas Instruments, USA.

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
