# Peer review of "A Low-Cost, Wireless, 3-D-Printed Custom Armband for sEMG Hand Gesture Recognition"

_sensors, 2019, doi:10.3390/s19122811_

Round 1

Reviewer 1 Report

This work presents the development of an armband for the capture of myoelectric signals combined with IMU, with 10 channels captured simultaneously, wirelessly, connected to a central station. The details on the design and architecture of the system were duly explained. The system was compared with the commercial system Myo Armband, using two databases, captured simultaneously, with 11 different gestures, including the neutral position (rest). Three different gesture recognition systems were trained, combining input data (characteristics, raw data and spectrograms) with two classifiers used in the literature (LDA and ConvNet). The results showed a better performance of 3DC compared with Myo Armband.

The main contribution of this work seems to be the presentation of the development of a new hardware system, which can be considered as an alternative with characteristics similar or superior to those of the state of the art. Validation of the system with a database has been proposed to demonstrate the performance of the proposed system.

I suggest attention to the following details, in order to improve the quality of work:

-          Details were not included about the base station that receives the system data. This system is configurated in a notebook?

-          The data acquisition protocol section does not describe what was the method used for the command to perform each movement. It can be assumed that a cue was generated as a stimulus to indicate the start and the transition between gestures. How can the authors deal with the fact that the response of a movement to the sent cue (every 5 seconds) contains a delay due to the human reaction?

Some studies have reported a delay up to almost a second. Some methods to deal with the drawback to separate the data from each gesture has been described in the literature. The techniques for onset and offset detection are widely used in several works. Possibly, this factor has influenced the accuracy of the classification described in the manuscript, because the training/validation may contain data belonging to another class of movements different from those to be classified.

-          The parameters defined for system validation, such as latency, allow processing for real-time classification. The text suggests that the processing of the data was done offline, due to the results include the comparison of different classifiers. However, the processing on the “offline” mode could be explicitly detailed in the manuscript.

-          Accuracy is a global measure and does not show the classification rate of individual tasks. One or more more robust measure of performance could be included for the comparison of the classifiers, in addition to the accuracy, which allows demonstrating the classification rate of each task, such as specificity, true- and false-positive rates, among others.

Other minor aspects to be taken into account are described below:

In the legend of Figure 1, in the second line, space is missed: "The etiquette..."

The Fig. 3(b) were not be cited in the text.

It would be kindly better if Fig. (b) would be above Fig. (c).

Line 165. The cited figure mentioned in the text is Fig. 3. Instead, it should be Fig. 3(c).

Line 172. Deleted the repeated word "the"

The x-axis label of Figure 8 should be translated into English

According to the data acquisition protocol (lines 215 - 216), the participants hold each gesture for five seconds. However, the segments of recording signals showed in Fig 8 seems to be smaller than 5 seconds for each gesture. Please, be more clear about it

Reviewer 2 Report

This work presents a 3D printed wireless-armband, within the Myo Armband range price; although featuring 10-channel, 1000sps, dry-electrode, and a 9-axis inertial measurement unit. The presented system is compared with the Myo Armband by employing a new offline dataset for 22 able-bodied participants performing eleven hand/wrist. Results shows that The proposed system outperforms the Myo Armband.

The article is well-structured and interesting. Reading very coherent and accessible for readers to adapt to the concepts. It lacked the inertial measurement unit fundamentals and the implementation information, at least at the same detail other information is provided.

Inertial measurement unit (IMU) devices measure acceleration, angular rates, and magnetic field vectors in their own three-dimensional local coordinate system, and, with proper calibration, represent an orthonormal base that is aligned with the outer sensor casing [doi.org/10.3390/app8112032]. Some commercial devices provide the orientation estimation of this sensor for a global fixed coordinate system. They do this by using algorithms that provide orientation information in the form of quaternion, rotation matrix, or Euler angles. These algorithms employ a strap-down integration of the angular rates to obtain a first estimate of the orientation.

To present a complete picture of the system, authors should include the IMU general fundamentals, pertinent references, a description of the measured variables, the calculated variables, and the explanation to understand the relation between IMU and EMG signals to identify hand/wrist gestures.

Reviewer 3 Report

The authors have developed a low-cost armband for sEMG data acquisition (3DC Armband), to overcome the limitations of current technology.  They also made available de comparison data set, that includes the 3DC Armband and the popular Myo Armband.  The manuscript is well written, and it was an enjoyable read.  I have a few comments to help the authors improve the quality of the manuscript.

·        In the comparison between Myo and 3DC Armbands, why did you use Wilcoxon Signed Rank test for all comparisons? Instead, indices are tested for normality and t-test is used when data holds normality, otherwise Wilcoxon Signed Rank test is used.  Elaborate on our approach.

·        In your approach, each participant is considered as a separated dataset, meaning that your final application is for an Armband that needs to be calibrated for each specific subject before used.  On the other hand, you could have used a subject-independent validation approach, in which you leave one subject out each time, and train your model using the remaining 22 subjects.  Accuracy is computed in the data of the subject left out. In this way you have 22 accuracies and the average accuracy will be independent of the subjects. In the real world, you might want to deploy a device that can work in a subject’s arm whose data is not available for training (e.g. an amputee). Did you consider this approach?

·        I don’t understand why the authors conclude that the 3DC Armband requires more warm-up time than the Myo Armband.  The 3DC Armband outperformed the Myo Armband in almost all scenarios, except by the one using only one cycle.  However, in such scenario the Myo Armband did not outperform the 3DC Armband, so it cannot be concluded than the Myo Armband requires less warm-up time. 

·        Also, it is not clear how the authors link the warm-up time required with the size of the electrodes.  It is more related to the material with which the electrodes are made. 

·        By the way, did you or some other group characterized the ENIG electrodes before (electrode-skin contact impedance spectroscopy, electrode equivalent model, etc.)?

·        Detailed construction details and files for building the 3DC Armband will be provided for public replication?

·        Although is a common term, consider define SoC the first time used in the manuscript.

Round 2

Reviewer 2 Report

The manuscript has been significantly improved and now can be considered for publication in Sensors. The authors included a new subsection, adding the Inertial Measurement Unit information.

Reviewer 3 Report

The authors have addressed all my comments and suggestions.